# QuaRL: Quantization for Fast and Environmentally Sustainable Reinforcement Learning

**Srivatsan Krishnan**[*]                                                                      *srivatsan@seas.harvard.edu*
*Harvard University*

**Maximilian Lam**[*]                                                                              *maxlam@g.harvard.edu*
*Harvard University*

**Sharad Chitlangia**[*†]                                                                      *chitshar@amazon.com*
*Amazon Advertising*

**Zishen Wan**[‡]                                                                                      *zishenwan@gatech.edu*
*Georgia Institute of Technology*

**Gaberial Barth-Maron**                                                                      *gabrielbm@google.com*
*DeepMind*

**Aleksandra Faust**                                                                              *sandrafaust@google.com*
*Google Research, Brain*

**Vijay Janapa Reddi**                                                                              *vj@eecs.harvard.edu*
*Harvard University*

**Reviewed on OpenReview:** *https://openreview.net/forum?id=xwWsiFmUEs*

## Abstract

Deep reinforcement learning continues to show tremendous potential in achieving task-level autonomy, however, its computational and energy demands remain prohibitively high. In this paper, we tackle this problem by applying quantization to reinforcement learning. To that end, we introduce a novel Reinforcement Learning (RL) training paradigm, *ActorQ*, to speed up actor-learner distributed RL training. *ActorQ* leverages 8-bit quantized actors to speed up data collection without affecting learning convergence. Our quantized distributed RL training system, *ActorQ*, demonstrates end-to-end speedups between $1.5 \times$ and $5.41\times$, and faster convergence over full precision training on a range of tasks (Deepmind Control Suite) and different RL algorithms (D4PG, DQN). Furthermore, we compare the carbon emissions (Kgs of CO2) of *ActorQ* versus standard reinforcement learning algorithms on various tasks. Across various settings, we show that *ActorQ* enables more environmentally friendly reinforcement learning by achieving carbon emission improvements between $1.9\times$ and $3.76\times$ compared to training RL-agents in full-precision. We believe that this is the first of many future works on enabling computationally energy-efficient and

---

[*]Equal Contribution
[†]Work done when Sharad was a visiting undergraduate student at Harvard.
[‡]Work done when Zishen was a graduate student at Harvard.

sustainable reinforcement learning. The source code is available here for the public to use: https://github.com/harvard-edge/QuaRL.

## 1 Introduction

Deep reinforcement learning has attained significant achievements in various fields (Bellemare et al., 2013; Kempka et al., 2016; Kalashnikov et al., 2018; Silver et al., 2016; 2017; OpenAI, 2018; Chiang et al., 2019; OpenAI et al., 2019). Despite its promise, one of its limiting factors is long training times, and the current approach to speed up RL training on complex and difficult tasks involves distributed training (Espeholt et al., 2019; Nair et al., 2015; Babaeizadeh et al., 2016). Although distributed RL training has demonstrated significant potential in reducing training times (Hoffman et al., 2020; Espeholt et al., 2018), this approach also leads to increased energy consumption and greater carbon emissions. Recently, work by (Wu et al., 2021) indicates that improving hardware utilization using quantization and other performance optimizations can reduce the carbon footprint of training and inference of large recommendation models by 20% every six months. In this same vein, we believe reinforcement learning can also benefit from these optimization techniques, particularly quantization, to reduce training time, improve hardware utilization and and reduce carbon emissions.

In this paper, we tackle the following research question – *How can we speed up RL training without significantly increasing its carbon emissions?* To systematically tackle this problem, we introduce ActorQ. In ActorQ, we use quantization to reduce the computation and communication costs of various components of a distributed RL training system. Based on the characterization of core components of distributed RL training, we find that majority of the time is spent on actor policy inference, followed by the learner's gradient calculation, model update, and finally, the communication) cost between actors and learners (Figure 1). Thus, to obtain significant speedups and reduce carbon emissions, we first need to lower the overhead of performing actor inference. To achieve this in ActorQ, we employ quantized policy (neural network) inference, a simple yet effective optimization technique to lower neural network inference's compute and memory costs.

Applying quantization to RL is non-trivial and different from traditional neural network quantization. Due to the inherent feedback loop in RL between the agent and the environment, the errors made at one state might propagate to subsequent states, suggesting that actor's policies might be more challenging to quantize than traditional neural network applications. Despite significant research on quantization for neural networks in supervised learning, there is little prior work studying the effects of quantizing the actor's policy in reinforcement learning. To bridge this gap in literature, we benchmark the effects of quantized policy during rollouts in popular RL algorithms namely DQN (Mnih et al., 2013), PPO (Schulman et al., 2017), DDPG (Lillicrap et al., 2015), A2C (Mnih et al., 2016). Our benchmarking study shows that RL policies are resilient to quantization error, and the error does not propagate due to the feedback loop between the actor and the environment.

Although our benchmarking study shows the potential benefits of applying quantization (i.e., no error propagation), we still need to apply quantization during RL training to realize real-world speedups. Applying quantization during RL training may seem difficult due to the myriad of different learning algorithms (Lillicrap et al., 2015; Mnih et al., 2016; Barth-Maron et al., 2018) and the complexity of these optimization procedures. Luckily, most popular RL training algorithms can be formulated using the actor-learner training paradigm such as Ape-X (Horgan et al., 2018) and ACME (Hoffman et al., 2020). ActorQ leverages the fact that many RL training procedures can be formulated in a general actor-learning paradigm in order to apply quantization specifically during actor's policy inference and policy broadcast between learners to actors during the RL training. To demonstrate the benefits of our approach, we choose two RL algorithms, namely DQN (Mnih

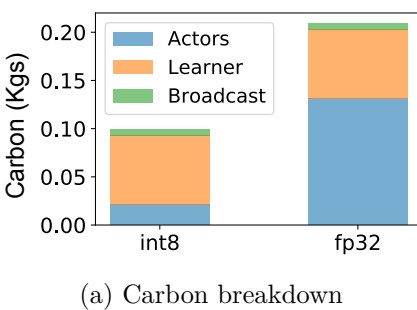

(a) Carbon breakdown

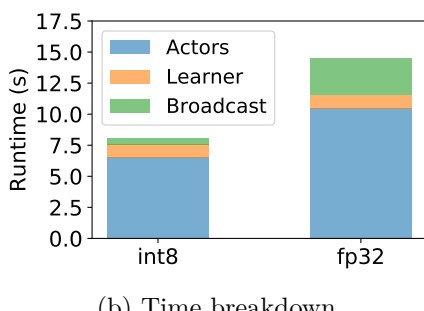

(b) Time breakdown

Figure 1: (a) Breakdown of carbon emissions between using a quantized and non-quantized policy in RL training for MountainCar. (b) Wall clock time for 'Actors', 'Learner', and 'Broadcast' for over a window of 1000 steps in ACME (Hoffman et al., 2020)

et al., 2013) and D4PG (Barth-Maron et al., 2018) from Deepmind's ACME framework (Hoffman et al., 2020) and show speedups in training time and reduced carbon emissions while maintaining convergence.

In summary, our fundamental contributions are as follows:

- We introduce *ActorQ*, to speed up distributed reinforcement learning training. *ActorQ* operates by quantizing the actor's policy, thereby speeding up experience collection. *ActorQ* achieves between $1.5 \times$ and $5.41\times$ speedup on a variety of tasks from the Deepmind control suite (Tassa et al., 2018) and OpenAI gym (Brockman et al., 2016) compared to its full-precision counterparts.

- Using our *ActorQ* framework, we further explore opportunities to identify various bottlenecks in distributed RL training. We show that quantization can also minimize communication overheads between actors and learners and reduce reinforcement learning time.

- Finally, by quantizing the policy weights and communication between actors and learners, we show a reduction in *carbon emissions* between $1.9 \times$ and $3.76\times$ versus full precision policies, thus paving the way towards sustainable reinforcement learning.

- To address lack of benchmarks in applying quantization for RL in the literature, we extensively benchmark quantized policies on standard tasks (Atari, Gym), algorithms (A2C, DQN, D4PG, PPO), and policy architecture (MLPs, CNNs). We demonstrate little to no loss in mean return, especially in the context of using quantized policy for rollouts.

## 2 Related Work

Both quantization and reinforcement learning in isolation have been the subject of much research in recent years. However, to the best of our knowledge, they have seldom been applied together (as surveyed in a broad range of quantization and RL papers) to improve RL efficiency. Below we provide an overview of related works in both quantization and reinforcement learning and discuss their contributions and significance in relation to our paper.

### 2.1 Quantization

Quantizing a neural network reduces the precision of neural network weights, reducing memory transfer times and enabling the use of fast low-precision compute operations. Innovations in both post-training quantization (Krishnamoorthi, 2018; Banner et al., 2018; Zhao et al., 2019; Tambe

| Metrics | A3C | CULE | Ray | Gorila | Seed-RL | ACME | Actor-Q (Our Work) |
|---|---|---|---|---|---|---|---|
| Method | Distributed | Distributed | Distributed | Distributed | Distributed | Distributed | Distributed/Standalone |
| Increase in Speed-up | ✓ | ✓ | ✓ | ✓ | ✓ | ✓ | ✓ |
| Decrease in Carbon Emission | ✗ | ✗ | ✗ | ✗ | ✗ | ✗ | ✓ |
| Decrease in Energy | ✗ | ✗ | ✗ | ✗ | ✗ | ✗ | ✓ |
| Decrease in Communication Cost | ✗ | ✗ | ✗ | ✗ | ✗ | ✗ | ✓ |
| Framework Agnostic | ✓ | ✓ | ✗ | ✗ | ✗ | ✗ | ✓ |

Table 1: Comparison of prior works on speeding-up RL training wrt to speed-up (lower training times), energy, and carbon emissions. Previous works compared include Nvidia's CULE (Dalton et al., 2019), Ray (Moritz et al., 2018), Gorila (Nair et al., 2015), Seed-RL (Espeholt et al., 2019) and ACME (Hoffman et al., 2020)

et al., 2020) and quantization aware training (Dong et al., 2019; Hubara et al., 2018; Choi et al., 2018) demonstrate that neural networks may be quantized to very low precision without accuracy loss, suggesting that quantization has immense potential for producing efficient deployable models. In the context of speeding up training, research has also shown that quantization can yield significant performance boosts. For example, prior work on half or mixed precision training (Sun et al., 2019; Das et al., 2018) demonstrates that using half-precision operators may significantly reduce compute and memory requirements while achieving adequate convergence.

Although much research has been conducted on quantization and machine learning, the primary targets of quantization are applications in the image classification and natural language processing domains. Quantization as applied to reinforcement learning has been absent in the literature.

## 2.2 Reinforcement Learning & Distributed Reinforcement Learning Training
Significant work on reinforcement learning range from training algorithms (Mnih et al., 2013; Levine et al., 2015) to environments (Brockman et al., 2016; Bellemare et al., 2013; Tassa et al., 2018) to systems improvements (Petrenko et al., 2020; Hoffman et al., 2020). From a system optimization perspective, reinforcement learning poses a unique opportunity compared to traditional supervised machine learning algorithms as training a policy is markedly different from standard neural network training involving pure backpropagation (as employed in learning image classification or language models). Notably, reinforcement learning training involves repeatedly executing policies on environments (experience generation), communicating across various components (in this case between devices performing rollouts and the main device updating the policy), and finally, learning a policy based on the training samples obtained from experience generation. Experience generation is trivially parallelizable by employing multiple devices that perform rollouts simultaneously, and various recent research in distributed and parallel reinforcement learning training (Kapturowski et al., 2018; Moritz et al., 2018; Nair et al., 2015) leverages this to accelerate training. One significant work is the Deepmind ACME reinforcement learning framework (Hoffman et al., 2020), which enables scalable training to many processors or nodes on a single machine.

## 3 ActorQ: Quantization for Reinforcement Learning
In this section, we introduce *ActorQ* a quantization method for improving the run time efficiency of actor-learner training. We first provide a high-level overview of the *ActorQ* system. Then, we

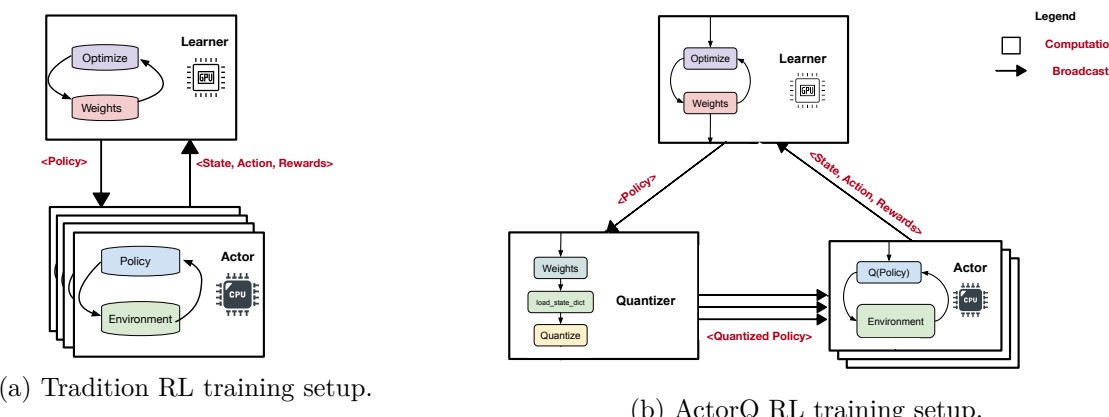

(a) Tradition RL training setup.

(b) ActorQ RL training setup.

Figure 2: (a) Traditional RL training setup. In a non-distributed RL training scenario, the number of actors is 1, and all the components are run on the same machine. In distributed RL training setup (e.g., ACME), the actors are distributed across multiple CPUs. (b) ActorQ system setup. In ActorQ, we add a quantizer block to the RL-training loop. The learner performs full-precision policy training in GPU. Before the policy is broadcasted to the actors, the quantizer block quantizes the policy. The quantized policy reduces the communication of the updated policy between the GPU learner and CPU actors. The actors run rollouts on the quantized policy for the experience generation. The learner and actors are instrumented with carbon monitoring APIs (Henderson et al., 2020) to quantify the impact of carbon emission with and without quantization.

characterize the effects of quantization on different reinforcement learning algorithms. Lastly, we apply quantization to a distributed RL training framework to show speed-ups on a real system. Our results demonstrate that apart from reducing training time, *ActorQ* also leads to lower carbon emissions, thus paving the way towards sustainable reinforcement learning research.

### 3.1  *ActorQ* System Architecture

Traditional RL training can be formulated within context of an actor- learner training paradigm (Horgan et al., 2018) as shown in Fig. 2a. In this framework, one or multiple actors collect new training examples based on its current policy and relays them to the learner (or replay buffer), which utilizes the generated experience to update the weights of its own policy. Periodically, the learner broadcasts its policy to the actors which update their internal policy with the learner's, synchronizing the models. In the non-distributed formulation, the actor, learner, and replay buffer reside in a single machine. However, in the case of the distributed RL, the actor, learner, and the replay buffer run on different compute nodes, constantly communicating with each other. We refer to this communication between the actor/learner components as broadcast.

*ActorQ* introduces quantization in the actor-learner reinforcement learning framework to speed up training. There are three main components in *ActorQ* system namely, 'Actors', 'Learner', and 'Quantizer' as shown in Figure 2. During training, each actor instance performs rollouts and initially uses a randomly initialized policy for decision making. At each step, the actors broadcast the environment state, action, and the reward for a given state to the learner. The learner uses this information to optimize the policy. Periodically, the learner broadcasts the updated policy to the actors, who then use the updated policy to perform future rollouts.

There are two main performance bottlenecks in reinforcement learning training. First, each actor uses a neural network policy to generate an action. Thus, how fast it can perform rollouts depends on the policy's inference latency. Second, the learner broadcasts the policy periodically to all the

actors. Broadcasting of the entire policy network to all actors can cause communication overheads and slow down training.

In ActorQ, we use quantization to reduce these bottlenecks and achieve end-to-end speed-up. To systematically integrate quantization into the training algorithm, we first perform a study to characterize the effects of applying post-training quantization and quantization aware training to various RL algorithms and environments.

In *ActorQ*, all actors use a quantized policy to perform rollouts. Additionally, the broadcasted policy is also quantized. Note that *ActorQ* maintains all learner computation in full precision to maintain learning convergence; further note that the learner is significantly faster than the actors due to the utilization of hardware acceleration (e.g., GPU) to perform batched updates during policy optimization (Espeholt et al., 2019). Meanwhile, actors are stuck performing rollouts that require repeated executions of single-example (non-batched) inference, and the difference in hardware utilization between the actors' rollouts and the learner's policy optimizations, based on our characterization, is one of the key areas leading to inefficiency. This motivates us to first apply quantization to the actors' inference and then to the policy broadcast (e.g: communication) between learners to actors.

While simple, *ActorQ* distinguishes from traditional quantized neural network training as the inference-only role of actors enables the use of low precision ($\leq 8$ bit) operators to speed up training. This is unlike traditional quantized neural network training, which must utilize more complex algorithms like loss scaling Das et al. (2018), specialized numerical representations Sun et al. (2019); Wang et al. (2018), stochastic rounding Wang et al. (2018) to attain convergence. This adds extra complexity and may also limit speedup and, in many cases, this is  still limited to half-precision operations due to convergence issues.

### 3.2   Effects of Quantization on Reinforcement Learning

As a first step towards developing *ActorQ*, we first perform experiments to benchmark the effects of quantization on RL. Insights gained from these experiments will help verify that quantization can be applied to learning without significantly degrading quality. To this end, we apply post-training quantization (PTQ) to just the policies of various reinforcement learning agents: we take an RL policy fully trained in fp32 (floating point precision) and apply post-training quantization to it, evaluating the impact of quantization on the resulting model. Performing this experiment allows us to understand whether the drift caused by quantization error at each policy step affects RL reward quality, and to what extent it impacts rewards with different degrees (i.e., number of bits) of quantization.

**Post-Training Quantization (PTQ)**
The post-training quantization is performed using standard uniform affine quantization (Krishnamoorthi, 2018) defined as follows:

$$Q_n(W) = round(\frac{W}{\delta})$$

where

$$\delta = \frac{|min(W, 0)| + |max(W, 0)|}{2^n}$$

Dequantization is defined as

$$D(W_q, \delta) = \delta(W_q)$$

| Algorithm → | A2C | | | | DQN | | | | PPO | | | | DDPG | | | |
|---|---|---|---|---|---|---|---|---|---|---|---|---|---|---|---|---|
| Datatype → | fp32 | fp16 | int8 | KL-div | fp32 | fp16 | int8 | KL-div | fp32 | fp16 | int8 | KL-div | fp32 | fp16 | int8 | KL-div |
| Environment ↓ | Rwd | Rwd | Rwd | | Rwd | Rwd | Rwd | | Rwd | Rwd | Rwd | | Rwd | Rwd | Rwd | |
| Breakout | 379 | 371 | 350 | 0.00262 | 214 | 217 | 78 | 0.06045 | 400 | 400 | 368 | 0.08892 | | | | |
| SpaceInvaders | 717 | 667 | 634 | 0.06066 | 586 | 625 | 509 | 0.01353 | 698 | 662 | 684 | 0.08115 | | | | |
| BeamRiders | 3087 | 3060 | 2793 | 0.00993 | 925 | 823 | 721 | 0.09787 | 1655 | 1820 | 1697 | 0.0186 | | | | |
| MsPacman | 1915 | 1915 | 2045 | 0.17536 | 1433 | 1429 | 2024 | 0.01531 | 1735 | 1735 | 1845 | 0.11978 | | | | |
| Qbert | 5002 | 5002 | 5611 | 0.01957 | 641 | 641 | 616 | 0.01995 | 15010 | 15010 | 14425 | 0.02573 | | | | |
| Seaquest | 782 | 756 | 753 | 0.03358 | 1709 | 1885 | 1582 | 0.02536 | 1782 | 1784 | 1795 | 0.02991 | | | | |
| Cartpole | 500 | 500 | 500 | 0.00113 | 500 | 500 | 500 | 0.1019 | 500 | 500 | 500 | 0.00566 | | | | |
| Pong | 20 | 20 | 19 | 0.01528 | 21 | 21 | 21 | 0.1257 | 20 | 20 | 20 | 0.01 | | | | |
| Walker2D | 399 | 422 | 442 | 0.05371 | | | | | 2274 | 2273 | 2268 | 0.03 | 1890 | 1929 | 1866 | 0.0376 |
| HalfCheetah | 2199 | 2215 | 2208 | 0.13427 | | | | | 3026 | 3062 | 3080 | 0.02 | 2553 | 2551 | 2473 | 0.0763 |
| BipedalWalker | 230 | 240 | 226 | 0.0252 | | | | | 304 | 280 | 291 | 0.03 | 98 | 90 | 83 | 0.0134 |
| MountainCar | 94 | 94 | 94 | 0.03705 | | | | | 92 | 92 | 92 | 0.07 | 92 | 92 | 92 | 0.0651 |

Table 2: Post-training quantization error for DQN, DDPG, PPO, and A2C algorithm on Atari and Gym. Quantization down to int8 yields similar episodic rewards to full precision baseline. To measure the policy distribution shift, we use KL-divergence. We measure KL-divergence between the fp32 policy and the int8 policy. The action distribution for the fp32 policy and int8 is shown in the appendix.

In our study, policies are trained in standard full precision. Once trained, we quantize them to fp16 and int8 first and dequantization them to simulate quantization error. For convolutional neural networks, we use per-channel quantization, which applies $Q_n$ to each channel of convolutions individually. Also, all layers of the policy are quantized to the same precision level.

We apply the PTQ to Atari arcade learning (Bellemare et al., 2013), OpenAI gym environments (Brockman et al., 2016) and different RL algorithms namely A2C (Mnih et al., 2016), DQN (Mnih et al., 2013), PPO (Schulman et al., 2017), and DDPG (Lillicrap et al., 2015). We train a three-layer convolutional neural network for all Atari arcade learning. For openAI Gym environments, we train neural networks with two hidden layers of size 64. In PTQ, unless otherwise noted, both weights and activations are quantized to the same precision.

Table 2 shows the rewards attained by policies quantized via post-training quantization in. The mean of int8 and fp16 relative errors ranges between 2% and 5% of the full precision model, which indicates that policies can be represented in 8/16 bits precision without much quality loss.

In a few cases (e.g., MsPacman for PPO), post-training quantization yields better scores than the full precision policy. We believe that quantization injected an amount of noise that was small enough to maintain a good policy and large enough to regularize model behavior; this supports some of the results seen by Louizos et al. (2018); Bishop (1995); Hirose et al. (2018).

To quantify the shift in policy distribution, we use KL-divergence (Kullback & Leibler, 1951). Table 2 shows the KL-divergence for the fp32 policy and int8 (quantized) policy. Across all the evaluated algorithms (both on-policy and off-policy), we observe that the KL-divergence is very small, suggesting that the quantization of policy does not change the inherent distribution significantly. The effect of small KL-divergence is also reflected in the minimal degradation in mean return.

We also visualize the action distribution of both the fp32 policy and int8. Fig. 3 shows the action distribution for two on-policy algorithms, namely A2C and PPO, for the Walker2D environment.

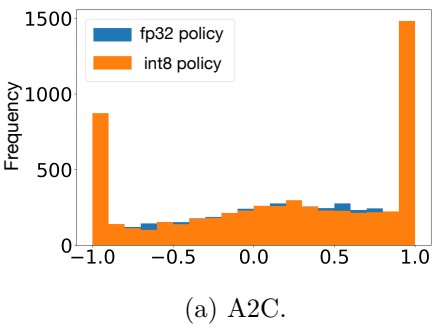

(a) A2C.

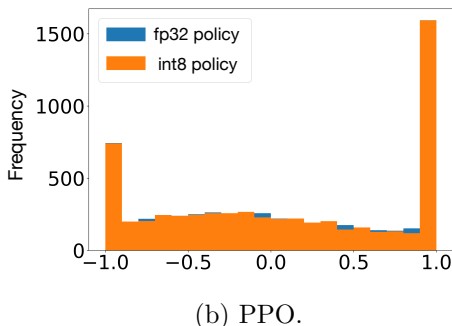

(b) PPO.

Figure 3: Small variation in action distribution for A2C and PPO in WalkerStand environment for fp32 and int8 policies for the same observation. We run each policy for 5000 steps. The small change in action distribution for quantized policy suggests safe exploration without significantly degrading the rewards.

We observe a small variation in the action distribution, suggesting that quantizing the policy allows the agent to perform a small safe exploration compared to the fp32 policy. This small variation is consistent with other environments and RL algorithms. Appendix C shows the action distributions for other environments.

Based on this study, we observe that the quantization of RL policy does not cause a significant loss in reward compared to an fp32 policy. Appendix A shows the results of applying quantization aware training (QAT) with RL. Since QAT uses fake quantization nodes to estimate the statistics of the parameter distributions, it allows for more aggressive quantizations. Our study on QAT suggests (See Appendix A) that we can safely quantize the policy up to 5-bits without significantly affecting the agent's rewards. However, to fully utilize the gains from aggressive quantization, we need native hardware support to run computations lower than 8-bits. Commonly used hardware accelerators used in RL training support native int8 computations. To that end, in ActorQ, we use PTQ for the quantizer block. Also, it is important to note that ActorQ uses simple uniform affine quantization to demonstrate how to apply quantization in RL policies; however, we can easily swap the quantizer function to include other quantization techniques (Krishnamoorthi, 2018).

In summary, at int8, we see minimal degradation in rewards. Hence, when designing quantizer block in *ActorQ*, we quantize the policy to int8. By using int8 quantization and leveraging the native int8 computation support in hardware, we achieve end-to-end speed-up in reinforcement learning training.

## 4 Experimental Setup

We evaluate the *ActorQ* system for speeding up distributed quantized reinforcement learning across various tasks in Deepmind Control Suite (Tassa et al., 2018). Overall, we show that: (1) we see significant speedup (between $1.5 \times$ and $5.41 \times$) in training reinforcement learning policies using *ActorQ*; (2) convergence is maintained even when actors perform int8 quantized inference; (3) Using *ActorQ*, we lower the carbon emissions between $1.9\times$ and $3.76\times$ compared to training without quantization.

### 4.1 *ActorQ Experimental Setup*

We evaluate *ActorQ* on a range of environments from the Deepmind Control Suite (Tassa et al., 2018). We choose the environments to cover a wide range of difficulties to determine the effects of

| Task | Algorithm | Steps Trained | Model Pull Frequency (Steps) |
|---|---|---:|---:|
| Cartpole Balance | D4PG | 40000 | 1000 |
| Walker Stand | D4PG | 40000 | 1000 |
| Hopper Stand | D4PG | 100000 | 1000 |
| Reacher Hard | D4PG | 70000 | 1000 |
| Cheetah Run | D4PG | 200000 | 1000 |
| Finger Spin | D4PG | 200000 | 1000 |
| Humanoid Stand | D4PG | 500000 | 100 |
| Humanoid Walk | D4PG | 700000 | 100 |
| Cartpole | DQN | 60000 | 1000 |
| Acrobot | DQN | 100000 | 1000 |
| MountainCar | DQN | 200000 | 100 |

Table 3: Tasks evaluated using *ActorQ* range from easy to difficult, along with the steps trained for corresponding tasks, with how frequently the model is pulled on the actor side.

quantization on both easy and difficult tasks. The difficulty of the Deepmind Control Suite tasks is determined by (Hoffman et al., 2020). Table 3 lists the environments we tested on with their corresponding difficulty and number of steps trained. Each episode has a maximum length of 1000 steps, so the maximum reward for each task is 1000 (though this may not always be attainable).

Policy architectures are fully connected networks with three hidden layers of size 2048. We apply a Gaussian noise layer to the output of the policy network on the actor to encourage exploration; sigma is uniformly assigned between 0 and 0.2 according to the actor being executed. On the learner side, the critic network is a three-layer hidden network with a hidden size of 512. We train policies using D4PG (Barth-Maron et al., 2018) on continuous control environments and DQN (Mnih et al., 2013) on discrete control environments. We chose D4PG as it was the best learning algorithm in (Tassa et al., 2018; Hoffman et al., 2020), and DQN is a widely used and standard reinforcement learning algorithm. An example submitted by an actor is sampled 16 times before being removed from the replay buffer (spi=16) (lower spi is typically better as it minimizes model staleness (Fedus et al., 2020)).

All the experiments are run in a distributed fashion to leverage multiple CPU cores and a GPU. A V100 GPU is used on the learner, while the actors are mapped to the CPU (1 core for each actor). We run each experiment and average over at least three runs to compute the running mean (window=10) of the aggregated runs.

## 4.2   Measuring Carbon Emissions

For measuring the carbon emission for the run, we use the `experiment-impact-tracker` proposed in prior JMLR work (Henderson et al., 2020).[1] We instrument the *ActorQ* system with carbon monitor APIs to measure the energy and carbon emissions for each training experiment in *ActorQ*.

To measure the improvement (i.e., lowering of carbon emissions) when using the int8 policy, we take the ratio of carbon emission when using the fp32 policy and int8 policy as:

---

[1]https://github.com/Breakend/experiment-impact-tracker

| Task | Mean Return | Time to Convergence (s) | | Speedup | Carbon Emissions (Kgs) | | Carbon Emission Improvement |
| --- | --- | --- | --- | --- | --- | --- | --- |
| | | fp32 | int8 | | fp32 $(Co2_{fp32})$ | int8 $(Co2_{int8})$ | $Co2_{fp32}/Co2_{int8}$ |
| Cartpole Balance | 941.22 | 870.91 | 279 | 3.12× | 0.359 | 0.15 | 2.39 |
| Walker Stand | 947.74 | 871.32 | 534.37 | 1.63× | 0.67 | 0.178 | 3.76 |
| Hopper Stand | 836.41 | 2660.41 | 1699.17 | 1.57× | 0.34 | 0.17 | 2.00 |
| Reacher Hard | 948.12 | 1597 | 875.34 | 1.82× | 0.35 | 0.18 | 1.94 |
| Cheetah Run | 732.31 | 2517.3 | 891.84 | 2.82× | 0.263 | 0.12 | 2.19 |
| Finger Spin | 810.32 | 3256.56 | 1065.52 | 3.06× | 0.361 | 0.19 | 1.90 |
| Humanoid Stand | 884.89 | 13964.92 | 9302.82 | 1.51× | 0.55 | 0.27 | 2.04 |
| Humanoid Walk | 649.91 | 17990.66 | 6223.35 | 2.89× | 0.56 | 0.278 | 2.01 |
| Cartpole (Gym) | 198.22 | 963.67 | 260.1 | 3.70× | 0.188 | 0.089 | 2.11 |
| Mountain Car (Gym) | -120.62 | 2861.8 | 1284.32 | 2.22× | 0.21 | 0.098 | 2.14 |
| Acrobot (Gym | -107.45 | 912.24 | 168.44 | 5.41× | 0.198 | 0.097 | 2.04 |

Table 4: *ActorQ* time, speedups, carbon emission improvements to 95% reward on select tasks from DeepMind Control Suite and Gym. 8 bit inference yields between 1.5 × and 5.41 × speedup over full precision training. Likewise, int8 policy inference yields between 1.9× and 3.76× less carbon emission compared to fp32 policy. We use D4PG on DeepMind Control Suite environments (non-gym), DQN on gym environments.

$$Co2_{Improvment} = \frac{CO2_{fp32}}{CO2_{int8}},$$

where, $Co2_{fp32}$ and $Co2_{int8}$ are the carbon emissions when using fp32 and int8 policy respectively.

## 5 Results

### 5.1 Speedup, Convergence, and Carbon Emissions

We focus our results on three main areas: end-to-end speed-ups for training, model convergence during training and environmental sustainability from a carbon emissions perspective.

**End-to-End Speedups.** We show end-to-end training speedups with *ActorQ* in Figure(s) 4 and 6. Across nearly all tasks, we see significant speedups with both 8-bit inference. Additionally, to improve readability, we estimate the 95% percentile of the maximum attained score by fp32 and measure time to this reward level for fp32, int8, and compute corresponding speedups. This is shown in Table 4. Note that Table 4 does not take into account cases where fp16 or int8 achieve a higher score than fp32.

**Convergence.** We show the episode reward versus total actor steps convergence plots using *ActorQ* in Figure(s) 5 and 6. Data shows that broadly, convergence is maintained even with 8-bit actors across both easy and difficult tasks. On Cheetah, Run and Reacher, Hard, 8-bit *ActorQ* achieve even slightly faster convergence, and we believe this may have happened as quantization introduces noise which could be seen as exploration.

**Carbon Emissions.** Table 4 also shows the carbon emissions for various task in openAI gym and Deepmind Control Suite. We compare the carbon emissions of a policy running in fp32 and

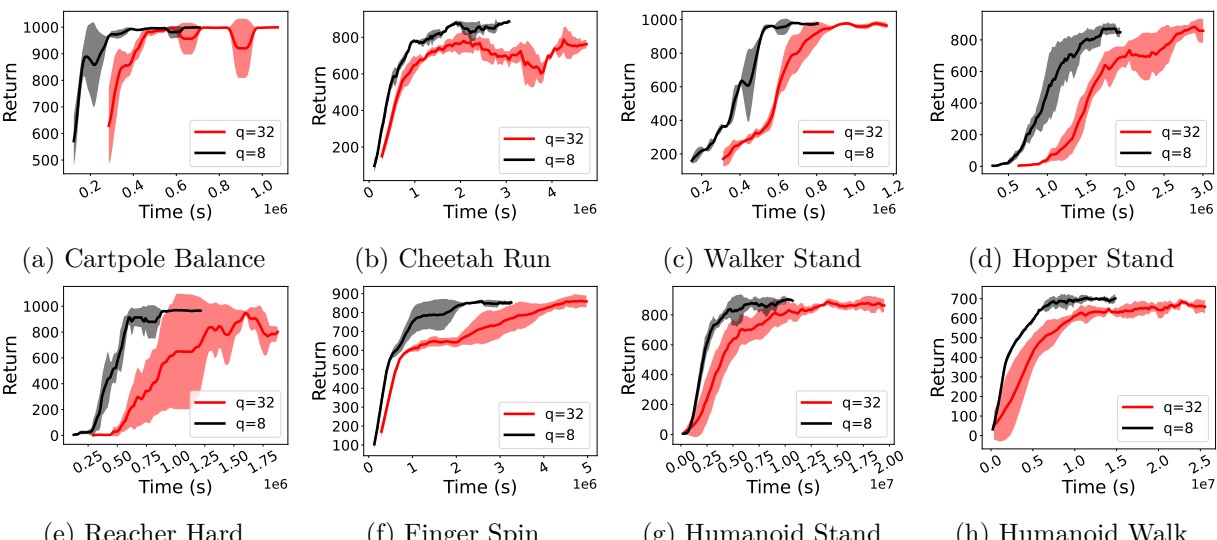

Figure 4: End-to-end speedups of *ActorQ* across various Deepmind Control Suite tasks using 8 bit and 32 bit inference. int8 training yields significant end-to-end training speedups over the fp32 baseline. The x-axis denotes the wall-clock time and y-axis denotes the reward. Training uses the D4PG algorithm.

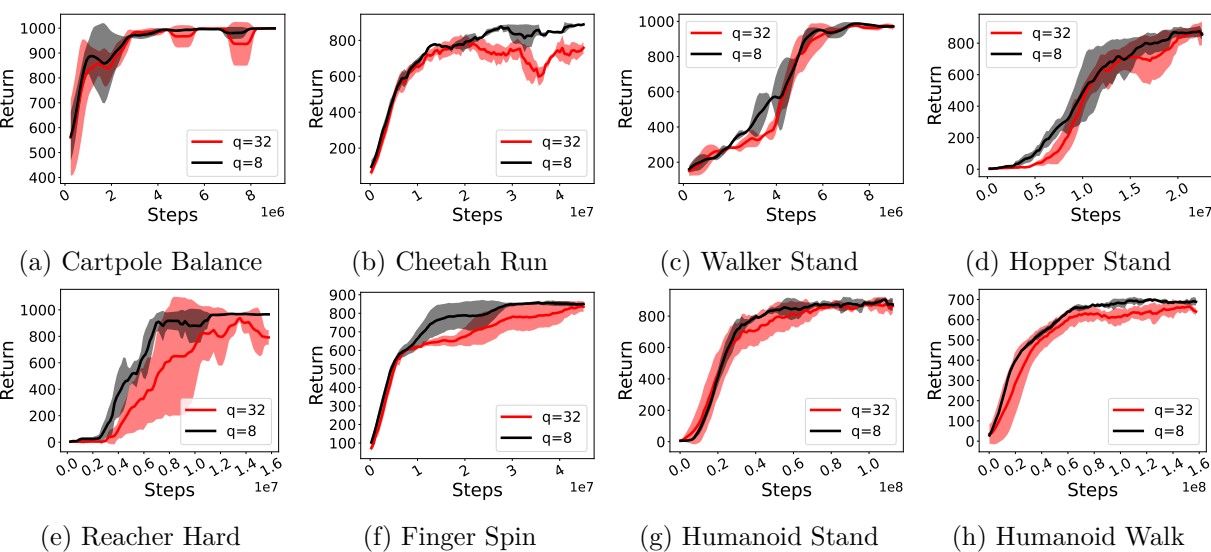

Figure 5: Convergence of *ActorQ* across various Deepmind Control Suite tasks using int8 and fp32 inference. int8 quantized training attains the same or better convergence than full precision training. Training uses the D4PG algorithm.

int8. We observe that quantization of policies reduces the carbon emissions anywhere from $1.9\times$ to $3.76\times$ depending upon the task. As RL systems are scaled to run on 1000's distributed CPU cores and accelerators (GPU/TPU), the absolute carbon reduction (measured in Kgs of CO2) can be significant.

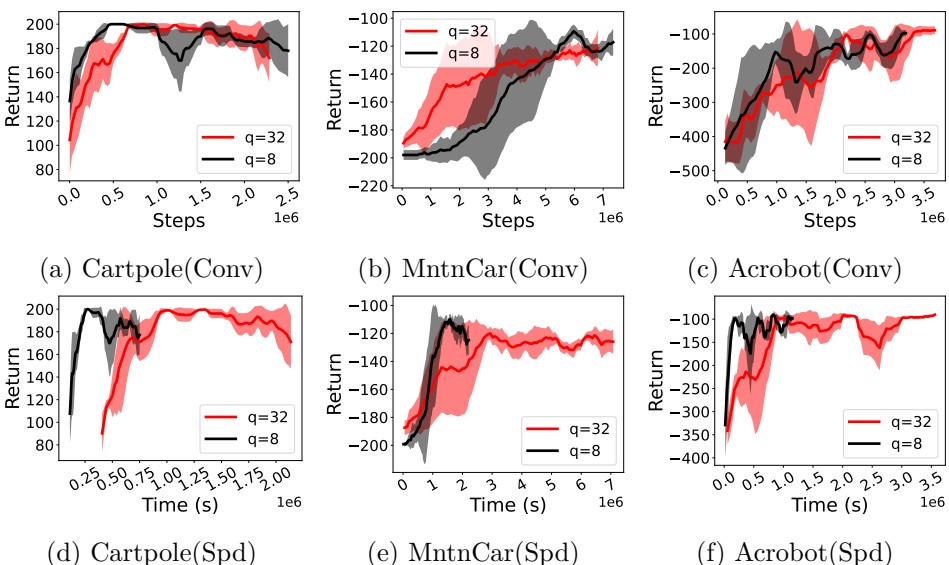

Figure 6: Convergence (Conv.) and end-to-end speedups (Spd) of *ActorQ* across various Gym tasks using int8 and fp32 inference. int8 training yields significant end-to-end training speedups over the full precision baseline. For the speed-up plots (d-f), the x-axis denotes wall-clock time. Training uses the DQN algorithm.

## 5.2 Communication vs Computation

The frequency of model pulls on actors is a hyperparameter and may have impacts on convergence as it affects the staleness of policies being used to populate the replay buffer; this has been witnessed in both prior research (Fedus et al., 2020) and our experiment with the hyperparameter. Figure 7a shows that a higher update frequency of 100 can help in faster convergence compared to an update frequency of 1000 for the Humanoid stand task. This hyperparameter has system-level implications since a higher update frequency can increase the communication cost (policy broadcast from learner to actors). In contrast, a lower update frequency can increase the computation cost since it will take more steps to converge, increasing the computation cost. Thus, to understand the tradeoff of quantization concerning this hyperparameter, we explore the effects of quantization of communication versus computation in both communication and computation-heavy setups.

To quantize communication, we quantize policy weights to int8 and compress them by packing them into a matrix, thus, reducing the memory of model broadcasts by $4\times$. Naturally, quantizing communication would be more beneficial in the communication heavy scenario, and quantizing compute would yield relatively more gains in the computation-heavy scenario.

Figure 8 shows an ablation plot of the gains of quantization on both communication and computation in a communication heavy scenario (frequency=30) versus a computation-heavy scenario (frequency=300).

The figures show that in a communication heavy scenario (Figure 8a), quantizing communication may yield up to 30% speedup; conversely, in a computation-heavy scenario (Figure 8b) quantizing communication has little impact as the overhead is dominated by computation. Therefore, we believe that communication would incur higher costs on a networked cluster as actors scale.

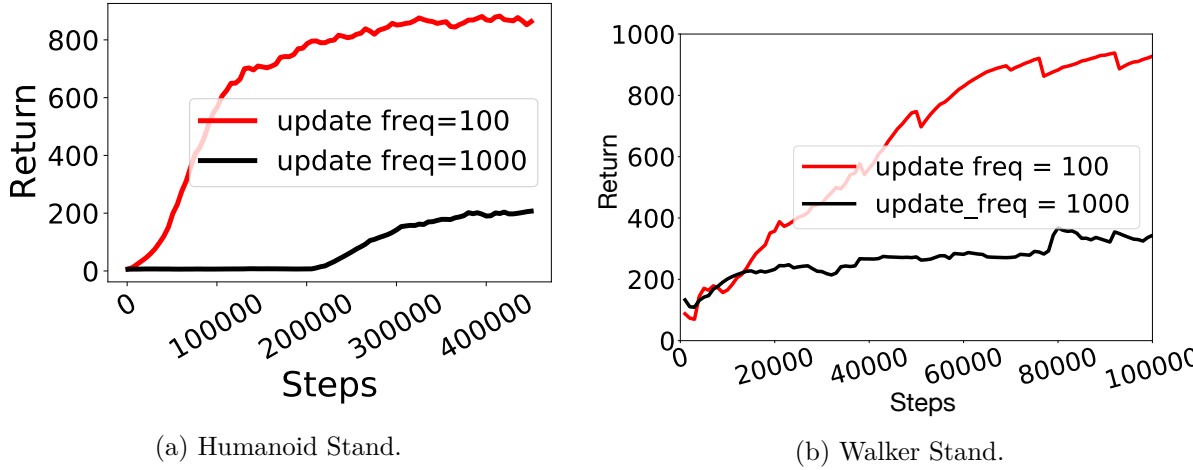

(a) Humanoid Stand.

(b) Walker Stand.

Figure 7: Studying the effect of training with more frequent actor pulls in Humanoid stand and Walker Stand. We observe that the more frequent actor pulls learns faster than with less frequent actor pulls and demonstrates model pull frequency affects staleness of actor policies and may have an effect on training.

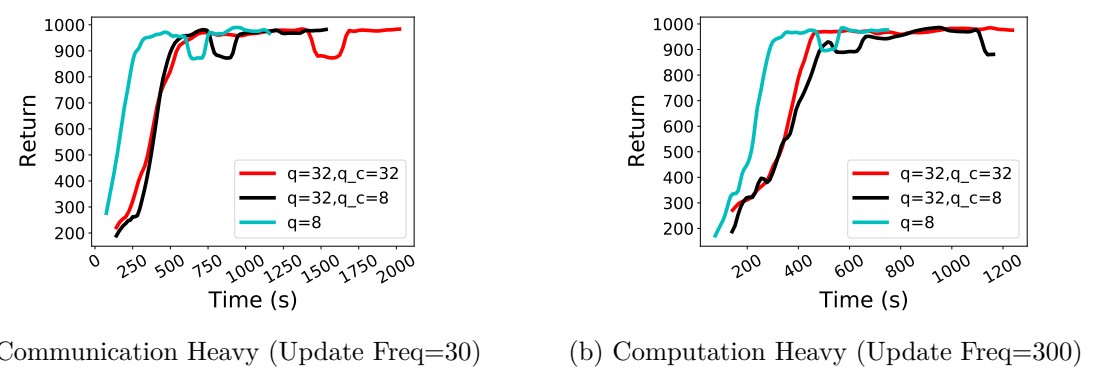

(a) Communication Heavy (Update Freq=30)

(b) Computation Heavy (Update Freq=300)

Figure 8: Effects of quantizing communication versus computation in compute heavy and communication heavy training scenarios. $q$ is the precision of inference; $q\_c$ is the precision of communication. Note q=8 implicitly quantizes communication to 8 bits. Experiment run on the walker stand task, using the D4PG algorithm.

### 5.3 Rationale for Why Quantization of Actors Speed-Up RL Training

We further break down the various components contributing to runtime on a single actor to understand how quantization of actor's policy inference speeds up training. Runtime components are broken down into: step time, pull time, deserialize time, and load_state_dict time. Step time is the time spent performing neural network policy inference. Pull time is the time between querying the Reverb queue (DeepMind, 2020) for a model and receiving the serialized models' weights; deserialize time is the time spent to deserialize the serialized model dictionary; `load_state_dict` time is the time to call PyTorch `load_state_dict` (used for loading and storing the policy).[2]

Figure 9a shows the relative breakdown of the component runtimes with fp32 (denoted as q=32), fp16 (denoted as q=16), and int8 (denoted as q=8) quantized inference in the communication heavy scenario. It is communication heavy because every 30 steps (Update Freq=30), the learner's policy is updated to each actors. This causes increased communication between the learner and actor

---

[2]https://bit.ly/pytorch_api

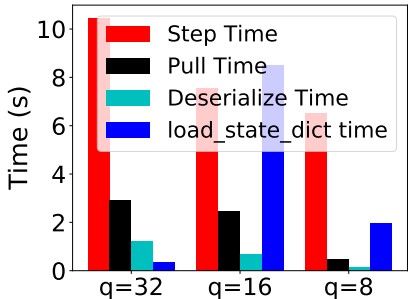

(a) Communication Heavy (Update Freq=30)

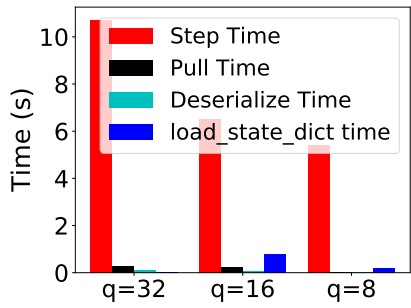

(b) Computation Heavy (Update Freq=300)

Figure 9: Breakdown of components for quantized and non-quantized training over 1000 steps.

components. As shown, step time is the main bottleneck, and quantization of the actor's policy significantly speed-up each roll-out, speeding up the overall training. Figure 9b shows the cost breakdown in the computation heavy scenario. While quantization speeds up the step time (Actor's policy inference), the pull time is also significantly reduced since there is less communication (Update Freq=300) happening between learner and actor. Quantization can still reduce the deserialize time and load_state_dict time since quantization lowers the memory footprint compared to the fp32 scenario.

In int8 and fp16 quantized training, the cost of PyTorch `load_state_dict` is significantly higher. An investigation shows that the cost of loading a quantized PyTorch model is spent repacking the weights from Python object into C data. Int8 weight repacking is noticeably faster than fp16 weight repacking due to fewer memory accesses. The cost of model loading suggests that additional speed gains can be achieved by serializing the packed C data structure and reducing the cost of weight packing.

## 6 Discussion & Future Work

To the best of our knowledge, our work is one of the first to experimentally and quantitatively demonstrated that quantization may be effectively applied to many facets of reinforcement learning, from obtaining high quality and efficient quantized policies, to reducing training times and eliminating carbon emissions. More specifically, we have shown that reinforcement learning policies may be quantized down to 4-5 bits (See Appendix A) without significantly affecting their performance; based on this result, we have developed a simple but effective method for speeding up reinforcement learning training, *ActorQ*, which achieves between 1.5× and 5.41× speedup over non quantized training, with a reduction in carbon emissions between 1.9× and 3.76×. In the future, alternative and more competitive methods are likely to emerge.

The computational requirements for RL training are growing (Espeholt et al., 2019) (Espeholt et al., 2018). Training OpenAI Five to play Dota 2 required a scaled-up version of Proximal Policy Optimization running on 512 GPUs and 51200 CPU cores (Berner et al., 2019). As we scale RL training to more thousands of cores and GPUs, even a 50% improvement as we have experimentally demonstrated (Table 4) will result in enormous savings in absolute dollar cost, energy, and carbon emissions. We believe that quantization, which is already a standard technique applied to non-reinforcement learning neural network models, will likewise be a critical technique in optimizing the performance of reinforcement learning policies. Our paper demonstrates that, like for standard neural networks, quantization yields significant benefits for reinforcement learning

while maintaining accuracy. We believe our work is a first step towards quantization being applied to reinforcement learning to achieve efficient, and environmentally sustainable training.

Several design decisions in our system warrant further discussion and research. In our design of the quantizer in *ActorQ*, we relied on simple uniform quantization, however, we believe other forms of aggressive quantization / compression (Park et al., 2016; Polino et al., 2018; Tambe et al., 2020; Lam et al., 2021) can also be applied (e.g., distillation, sparsification, etc.). Applying more aggressive quantization / compression methods may yield additional benefits to the performance / accuracy tradeoff obtained by the trained policies. Additionally, in order to achieve tangible speedups from quantization, the underlying hardware system must support quantized operations at the machine level. However, with increased hardware support for neural network execution, we believe that devices in the future will exhibit an increasing amount of hardware support for quantized operations (Jouppi et al., 2017). Finally, in *ActorQ*, we primarily focused on quantizing actor neural network execution and the communication between actors and learners (as these were the biggest computational bottlenecks), however, we believe that the learner's policy can similarly be quantized to achieve further performance benefits.

## 7 Conclusion

To the best of our knowledge, we are the first to apply quantization to not only speed up RL training and inference, but also reduce RL carbon emissions. We experimentally demonstrate that standard quantization methods can quantize policies down to $\leq 8$ bits with little quality loss. We present *ActorQ* to attain significant speedups and reductions in carbon emissions over full precision training. Our results demonstrate that quantization has considerable potential in speeding up both reinforcement learning inference and training while reducing carbon emissions. Future work includes extending the results to networked clusters to evaluate further the impacts of communication and applying quantization to reinforcement learning to different application scenarios such as the edge.

## Acknowledgements

The authors will like to thank Google for providing GCP research credits. We would also like to thank anonymous reviewers for their valuable feedback on the manuscript.

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

## Appendix

### A Quantization Aware Training (QAT)

To understand how aggressively (i.e., number of bits) we can quantify the RL policies, we use quantization aware training (QAT). In QAT, the RL policy weights and activations are passed through the quantization function $Q_n$ during inference; during backpropagation the straight-through estimator is used as the gradient of $Q_n$ (Krishnamoorthi, 2018)

$$\nabla_W Q_n(W) = I$$

Note that quantization aware training does not speed up training as all operations are still performed in floating-point. Quantization aware training is used primarily to train a model with simulated quantized weights and activations to evaluate the reward loss (if any) for a given RL task. Native hardware and library support for sub int8 quantization can speed policy inference time with quantized execution.

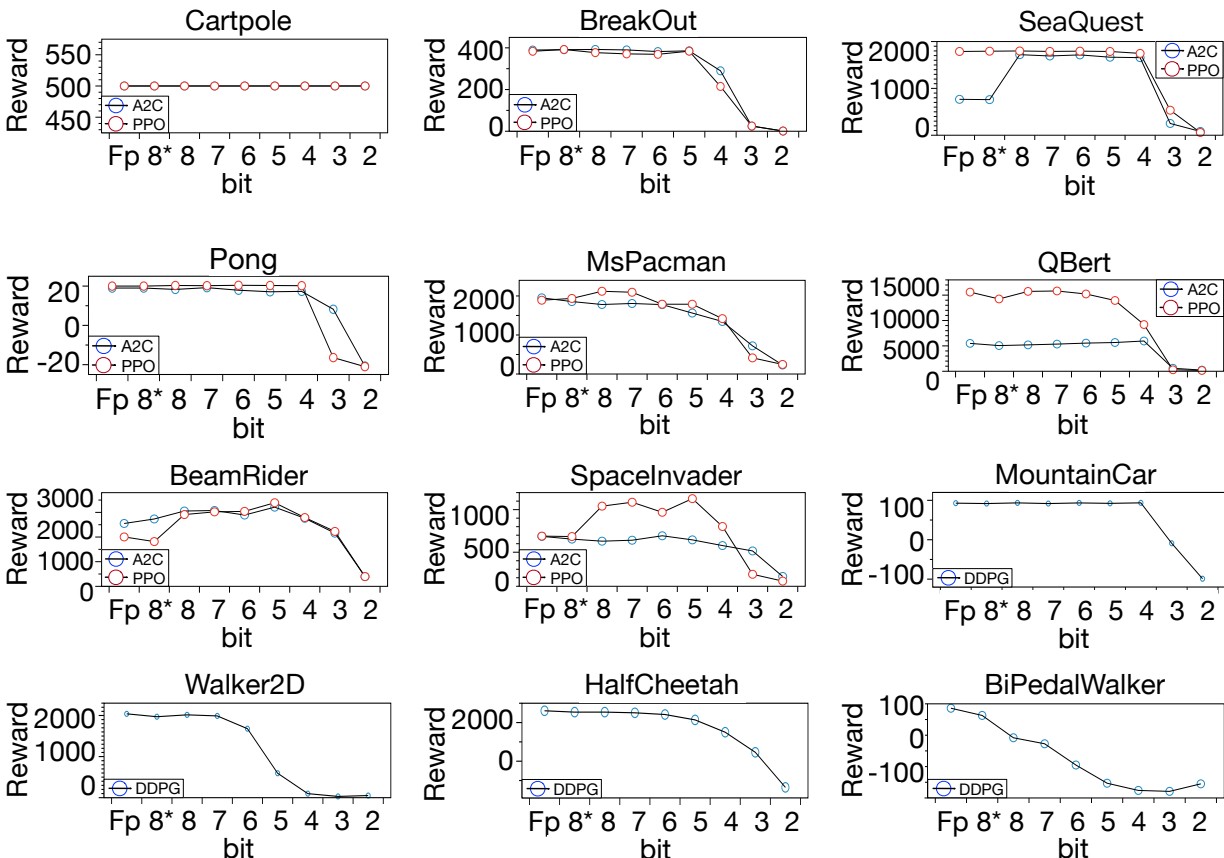

Figure 10: Quantization aware training of PPO, A2C, and DDPG algorithms on OpenAI gym, Atari, and PyBullet. FP denotes fp32 and 8* is achieved by int8 post-training quantization.

We present rewards for policies quantized via quantization aware training on multiple environments and training algorithms in Figure 10. We train a three-layer convolutional neural network for all Atari arcade learning. For openAI Gym environments, we train neural networks with two hidden layers of size 64. We train all agents for 10 Million steps. Since in QAT, the trainer inserts fake quantization node is the neural network graph (policy in the context of RL), and the statistics

of weights distribution during training are collected before the weights can be quantized. The quantization delay is a hyperparameter (Krishnamoorthi, 2018) that controls when the quantization is applied to the weights. One can consider this the "warm-up" period to collect the policy weight distribution statistics before applying quantization. We tried different values of quantization delay and used 5M since that gave us the best results.

Our results (see Fig. 10) suggest that generally, the performance relative to the full precision baseline is maintained until 5/6-bit quantization, after which there is a drop in reward, suggesting that for several of these environments, we can quantize the policy to 5-bits. However, to achieve measurable speed-up during training, there is a need for native hardware and library support (e.g., Nvidia Cuda/Intel MKL/ ARM Neon support in TensorFlow, Pytorch, and JAX frameworks).

Broadly, at 8-bits, we see no degradation in rewards. Hence, when applying quantization in *ActorQ*, we quantize the policy to 8-bits. By quantizing the policy at 8-bits and leveraging the native 8-bit computation support in hardware, we achieve end-to-end speed-up in reinforcement learning training.

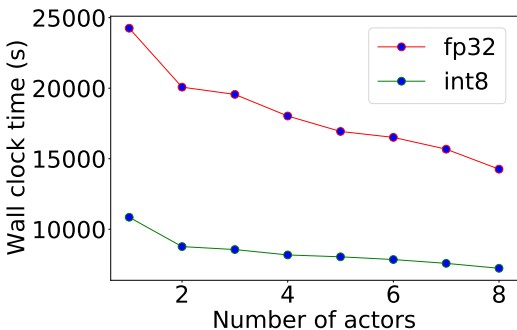

Figure 11: Wall clock time for training DQN with varying number of actors. The actor policies are trained in full precision (fp32) and int8. Benefits of quantization is agnostic to RL training setup and is effective for both single actor RL (non-parallel version) as well as distributed RL training (number of actors greater than one).

## B Parallelization and Scaling

In this paper, we show that applying quantization to reinforcement learning training improves training speed and lowers carbon emissions. Though in Table 1 explicitly compares with distributed RL training, we believe quantization can also improve single actor training.

The cost of RL training can be formulated as

$$C = c_b.n_a + c_a.n_a - c_o.n_p$$

Where , $n_a$: number of agents
$n_p$: number of processors
$c_b$ : cost for broadcasting (computed as a difference between quantized and non-quantized )
$c_a$: cost for actor inference (computed as a difference between quantized and non-quantized )
$c_o$: cost of overhead when more than one processor is used.

Our proposed method offers O(n) saving where n is the number of actors. Of course, this formula works only in the ideal cases, and will downgrade in real computer systems that run other processes

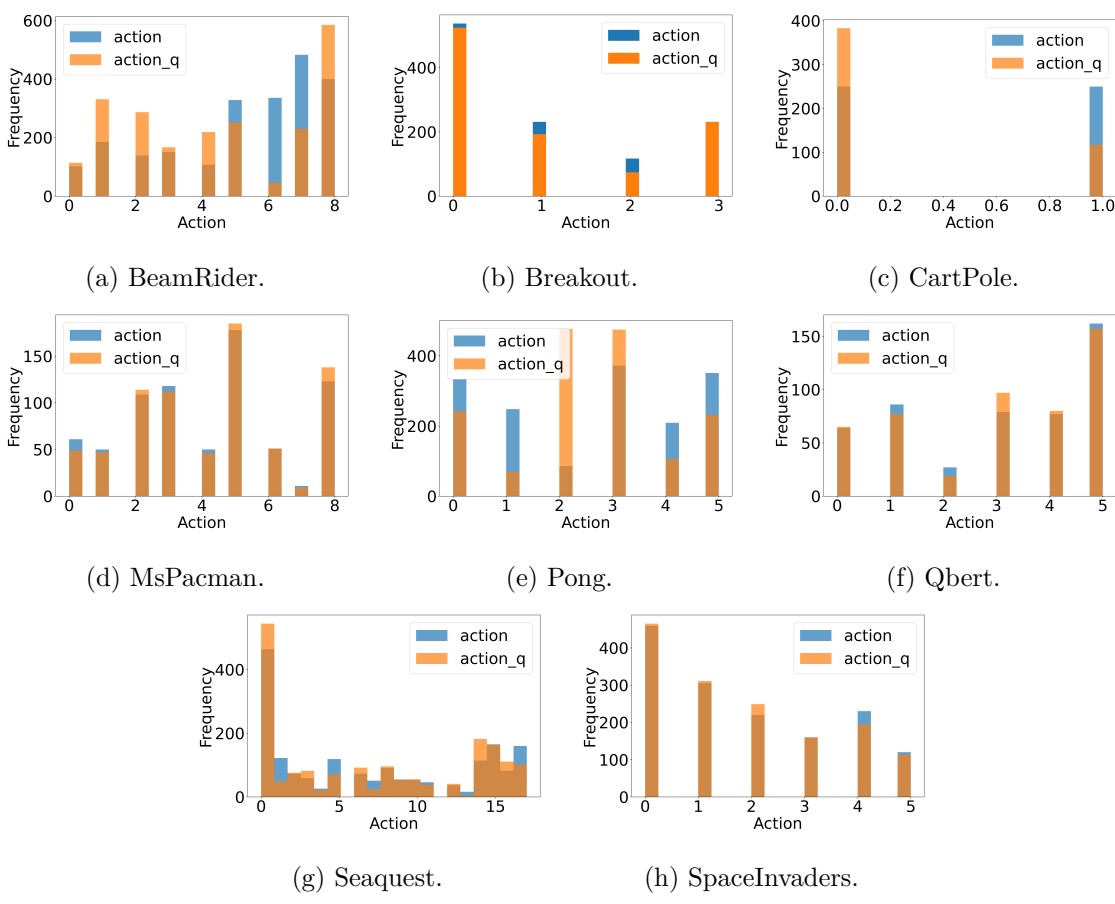

Figure 12: Variation in the action distribution for DQN between fp32 policy and quantized policy (int8) on several environments evalauted in Table 2. In the legends, 'action' and 'action_q'' corresponds to the actions taken by the fp32 and int8 policies.

and are subject to network traffic. Still these slowdowns are scaling with the number of processors, leaving overall linear improvement with different scalers depending on the number of processors used. Fig. 11 demonstrates the linear relationship between training time (savings) as number of actors scale. It is also important to note that in the case of single actor (non-distributed scenario), quantization of actor's policy still improves the training speed since $c_a$ is minimized due to quantized inference of actor's policy.

## C   Action Distribution Visualization

In this section, we include the variation in the action distribution between fp32 policy and quantized policy (int8) for PPO and DQN for several environments evaluated in Table 2.

**Methodology for visualizing action distributions.** To visualize the action distribution variation between int8 and fp32 policy, we load both the policies for rollouts. First, for each algorithm and task (environment) combination, we run a rollout of 5000 steps. The same observation is passed to the int8 and fp32 policies. Finally, the actions for both policies are logged and visualized.

Across different RL algorithms and tasks, we observe that the variation of the action between the fp32 policy and int8 policy is very small. Also, since the expected mean return between fp32 and int8 policies is very similar, it suggests that quantization of the policies facilitates safe exploration.

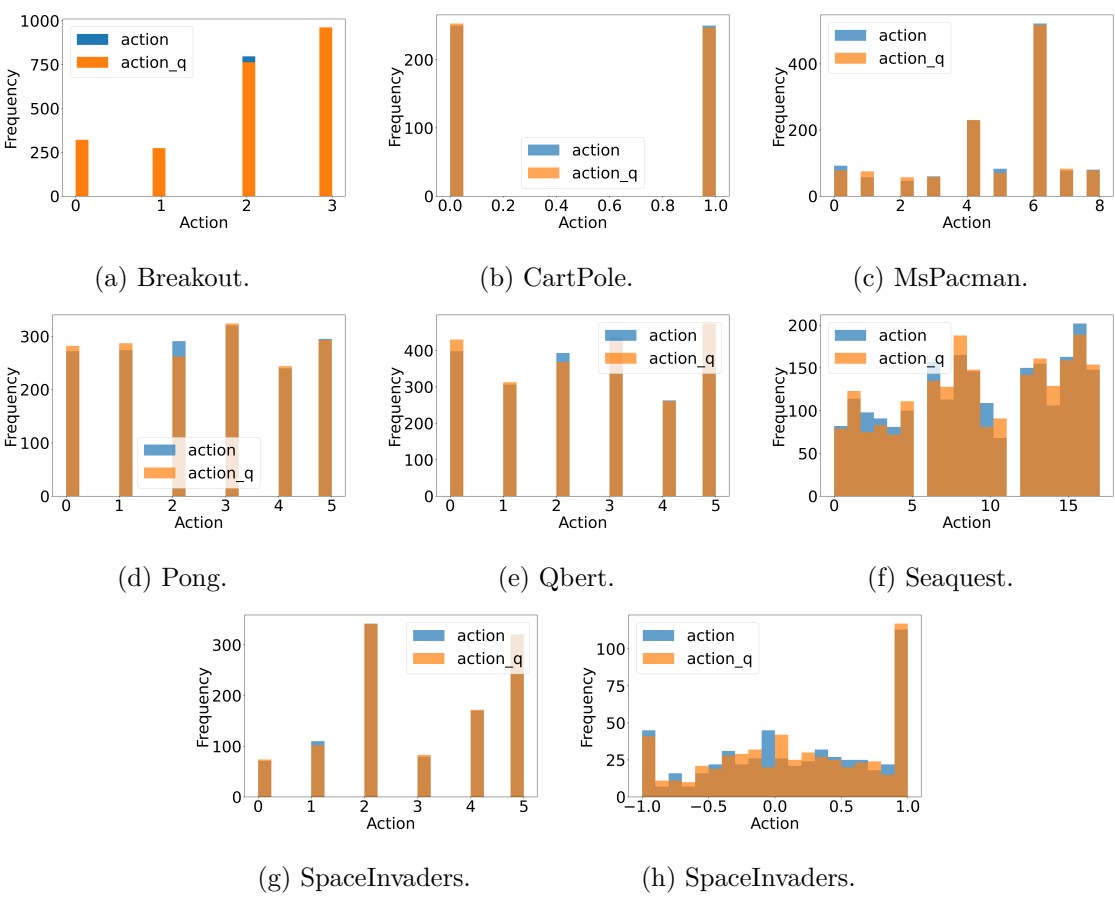

Figure 13: Variation in the action distribution for PPO between fp32 policy and quantized policy (int8) on several environments evalauted in Table 2. In the legends, 'action' and 'action_q'' corresponds to the actions taken by the fp32 and int8 policies.

Also, from a hardware utilization perspective, quantized computations (int8) are faster and more energy-efficient than fp32 computation. Hence, quantization is a simple yet effective strategy to improve the training speed of RL sustainably (i.e., lower carbon emissions).

