# OpenReview forum: "QuaRL: Quantization for Fast and Environmentally Sustainable Reinforcement Learning"
_TMLR — Accepted by TMLR_

### Review · Reviewer_axpH · 2022-04-26

**Summary Of Contributions:**

This paper studies how quantization of the weights of neural networks can speed up deep reinforcement learning. The quantized distributed RL training system, ActorQ is shown to have faster convergence on a range of tasks (Deepmind Control Suite). The paper also discussed the carbon emissions (Kgs of CO2) of ActorQ versus standard reinforcement learning on various tasks.

**Broader Impact Concerns:**

Some claims about "reducing training times and eliminating carbon emissions", "paving the path for sustainable reinforcement learning research" should be put in perspective with the broader context. Improving RL algorithms might not directly lead to these objectives and more careful claims and discussions would be useful (e.g. it can be used to just take up all the budget for more hyper-parameters tuning, which can be useful but would not lead to any reduction in carbon emission).

**Requested Changes:**

- The reason why there is an emphasis on parallelization in the paper (Table1, Section 2.2, etc.) is not perfectly clear. A lot of research is done to speed up RL training without requiring parallelization while still reducing training time/costs and this does not seem well explained. In Table 1, it is unclear why only parallelized algorithms have been chosen to compare with your contribution, while you specifically claim a reduction in energy/cost (which is not improved by parallelization).
- In the current form, it is difficult to understand clearly what the different elements of the algorithm are. What is the broadcasted policy (from Section 3.1) for instance in the context of DQN? The following is not trivial but is given without much explanation: "the learner is significantly faster than the actors due to utilizing the GPU (with actor policy inference achieving poor utilization on GPU due to operating over low batch sizes)." How is the following sentence supported by the results "Based on our characterization, we observe that time spent by actors to perform rollouts is significantly greater than that of learners"?


Other comments related to the writing that should be improved.
- typos/sentences not completed such as "and DDPG (Lillicrap et al., 2015) and."
- in Figure 2, the run time is shown to be about 15 seconds for "fp32", but what does the 15 seconds corresponds to? Figure 2 is only mentioned beginning of Page 2 but it is place beginning of page 5. Should it be brought closer to the relevant discussion?
- Table 3 is explained before Table 2 in the text and it makes things a bit difficult to follow.
- Figure 3 should be improved to increase clarity in what the figure is about. The y-axis is denoted as "frequency" but the results seem to provide a count on the number of times a given weight is encountered. The title "weight" looks as being at the same level than "DQN", "PPO", "A2C".
- Table 2, the term "reward" is not accurate I believe. Do you mean the experimental mean return (sum of rewards) over a trajectory?
- Referring to "the former" and "the latter" in Section 3.2 to refer from elements from the previous paragraph make things difficult to follow. Writing "Based on our analysis (see Figure 3)" is not ideal because Figure 3 (or even its caption) does not provide much of an "analysis".
- Things are not always carefully defined (e.g. "fp32")

**Strengths And Weaknesses:**

Strengths:
- The paper tackles an interesting problem. It considers many different base algorithms that are modified with quantization and a few different environments.

Weaknesses:
- The technical details are not always clearly explained. For instance, specific implementation details required for each base algorithm are not given.
- The writing and the presentation of results should be improved.

---

> ### Author Response · Authors · 2022-04-30
> **Reply to Reviewer axpH**
>
> We thank the reviewer for their detailed feedback.
>
> **[R]: Regarding emphasis on parallelization in the paper (Table1, Section 2.2, etc.) is not perfectly clear.**
>
> * Thank you so much of your inquiry. Reflecting on your question, we did a further analysis and realized that the total cost saving is:
>
>   C = c${_b}$ n${_a}$ + c${_a}$.n${_a}$ - c${_o}$ . n${_p}$
>
>  Where ,
>  n${_a}$:  number of agents
>
>  n${_p}$: number of processors,
>
>  c${_b}$: cost for broadcasting (computed as a difference between quantized and non-quantized )
>
>  c${_a}$: cost for actor inference (computed as a difference between quantized and non-quantized )
>
>  c${_o}$: cost of overhead when more than one processor is used.
>
> * Our proposed method offers O(n) saving where n is the number of agents. Of course, this formula works only in the ideal cases, and will downgrade in real computer systems that run other processes and are subject to network traffic. Still these slowdowns are scaling with the number of processors, leaving overall linear improvement with different scalers depending on the number of processors used.
>
> * In the case for a non-parallized version, the n${_a}$ is 1. However, quantization can still improve c${_a}$ and c${_b}$. Note that c${_a}$ and c${_b}$ depend only on the neural network size (policy architecture) and the processor, but not on the RL algorithm.
>
> * To verify our formula, we will empirically compute c${_a}$ and c${_q}$ for DQN. We will verify the analysis findings comparing the predicted and actual cost savings on 1, 2, 4, 6, and 8 processors. These experiments and validation will take 2-3 weeks to run. We will update the paper with the analysis and the empirical results.
>
>
> **[R] In the current form, it is difficult to understand clearly what the different elements of the algorithm are. What is the broadcasted policy (from Section 3.1) for instance in the context of DQN?**
>
> * We would like to clarify that we are not changing the inner functioning of these RL algorithms. We only change the precision of the actor policy, and evaluate its effect and show benefits in terms of speed-up and lower carbon emissions. In the standard on-line RL setting there is an actor / critic function that is currently being updated (learner), and another that collects the new samples (actor). For example, in DDQN, while one DQN is being updates, another collects the data. Every once in a while, the actor / critic function that is collecting the data in the replay buffer is updated. In the case of the distributed RL, the update consists of copying the actor / critic to different nodes. We refer to the update and copy as a broadcast. We will make this clarification in the paper.
>
> * We would be happy to modify Fig. 1 and add clarifications on how distributed RL training infrastructure works. For instance, frameworks like ACME and Seed-RL, the actor, learners and replay buffers are distributed on different machines for scalability. Broadcasting of the policy is when the learner updates the actor’s policy running on several different machines.
>
> **[R]: The following is not trivial but is given without much explanation: "the learner is significantly faster than the actors due to utilizing the GPU (with actor policy inference achieving poor utilization on GPU due to operating over low batch sizes).**
>
> * Learner samples trajectories from a replay buffer to learn a policy. This batching improves the accelerator hardware utilization (Espeholt et al., 2019b). We will add the necessary references to back the statement in our revised version.
>
> **[R]: In Figure 2, the run time is shown to be about 15 seconds for "fp32", but what does the 15 seconds corresponds to? Figure 2 is only mentioned beginning of Page 2 but it is place beginning of page 5. Should it be brought closer to the relevant discussion?**
>
> * In distributed training, there are three main components: Actors (performs inference of policy to take actions), learners (samples trajectories and to learn a policy), and broadcasts (communication of new policy from learners to actors). The actor runtime represents the wall clock time for a single actor to perform a single rollout, the learner runtime is the wall clock time it takes the GPU learner to perform backpropagation over a batch of training examples, and the communication time is the wall clock time aggregate of sending, deserializing and loading model data between the actor and learner.
> * The y-axis corresponds to the wall clock time. For actors, the time corresponds to running the policy to generate actions. For learners this time corresponds to the update of the policy gradients and for broadcast, this is the communication cost for transferring the new policy to each actor.
> * We will update the paper to move the figure closer to the relevant discussion.
> **[R]: Other typos, naming conventions (frequency, reward etc).**
> * We thank the reviewers for this feedback. We will fix these in the revised version.

---

> > ### Comment · Reviewer_axpH · 2022-05-28
> > **Many elements from the paper are still unclear**
> >
> > Many elements from the paper are still unclear. Among others, here are three elements:
> > - In the abstract, how can the speedups be between 1.5 × to 5.41×, while achieving (seemingly exactly) 2.8× less carbon emission and energy compared to training RL-agents in full-precision? Why do the authors give such a large uncertainty for the speedup but not for the carbon emission?
> > -  In the abstract, two different RL algorithms are mentioned (D4PG, DQN), in the end of the introduction there are 5 mentioned (A2C, DDPG, DQN, D4PG, PPO). Why is there such a difference between abstract and introduction?
> > - The paper is unclear about the benchmarks used and where they are used. For instance, first bullet point in the introduction: "ActorQ achieves between 1.5× and 2.5× speedup on a variety of tasks from the Deepmind control suite (Tassa et al., 2018) compared to its full-precision counterparts.", while the last bullet point of the introduction says "we extensively benchmark quantized policies on standard tasks (Atari, Gym)".

---

> > > ### Author Response · Authors · 2022-05-29
> > > **Response to many elements from the paper still not clear**
> > >
> > > We thank the reviewer for these questions. Upon reflecting on your questions, we agree that the distinction between ActorQ and benchmarking studies is not very obvious in the introduction and abstract. We have added clarifications in the revised version of the paper.
> > >
> > > **[R]: In the abstract, two different RL algorithms are mentioned (D4PG, DQN), in the end of the introduction, there are 5 mentioned (A2C, DDPG, DQN, D4PG, PPO). Why is there such a difference between abstract and introduction?**
> > > * The paper has two studies:
> > >     * **ActorQ.** System to demonstrate the real speed-ups and carbon emission saving by employing quantization during RL training. We    demonstrate it with D4PG and DQN across the Deepmind control suite and gym environments. Since ACME provides a similar interface for other RL algorithms, ideally one can extend ActorQ to other RL algorithms (beyond DQN and D4PG) available in ACME.
> > >
> > >   * **Benchmarking (Section 3.2)**. This study aims to understand the effects of quantization in RL in a broader context and use these learnings to design the quantizer block used in the ActorQ system. Here we take pre-trained RL policies (fp32 precision) trained using A2C, DDPG, DQN, and PPO for popular RL tasks and study if there is degradation in mean return for rollouts if we quantize these policies to int8. We have also expanded this benchmarking study in our subsequent revisions to include KL-divergence metrics requested by Reviewer 3t9v. These results can be found in Section 3.2 and Appendix C.
> > >
> > >  * Thus benchmarking studies show the potential of using quantization in RL. In contrast, ActorQ realizes that potential (i.e., speed-ups and carbon emission) by implementing quantization in a scalable distributed RL system.
> > >
> > > **[R]: Why do the authors give such a large uncertainty for the speedup but not for the carbon emission?**
> > > * We want to clarify that it is not an uncertainty measurement but a range of observed speedups for different environments/algorithms combinations from Table 4. We also want to clarify that the carbon emission number is not fixed and varies depending upon the task as reported in Table 4.
> > >
> > > * You are right that adding a range for carbon emissions similar to speedup would make it clearer. There wasn’t a particular reason why we did not include that range for carbon emissions in the abstract since Table 4 captures the range.
> > >
> > >  * We have fixed this to include that range in the abstract from Table 4 for carbon emissions in the current revisions.

---

### Review · Reviewer_nimW · 2022-05-01

**Summary Of Contributions:**

The authors propose to use quantization in distributed reinforcement learning. They show that significant speed up and reduced carbon emission can be obtained by quantizing the policy used for collecting data, without quantizing the learner part.

**Broader Impact Concerns:**

There is no broader impact statement, but it would be easy to write a positive one, putting forward the concern for carbon emission due to growing research in ever more complicated environments in RL.

**Requested Changes:**


Figure 1 shows ‘Actors’, a ‘Learner’, and a ‘Quantizer’, but Fig.2 studies the impact of the Actor, Learner and Broadcast parts. Is there a way to be more consistent? E.g. can the quantizer be considered as responsible for broadcast?

Up to section 3.2, the intent of the paper is rather clear, but then the distinction between post-training quantization (PTQ) and quantiza-
tion aware training (QAT) blurs the message by introducing several unclarities.

First, PTQ and QAT are presented as "two types of quantization", but it is unclear whether they are alternative quantization methods or if they can be combined. Second, what is described as QAT seems to only correspond to a study of the impact of the hardness of the quantization, thus I don't unsderstand the name "quantization-aware training". From Fig. 1, I rather understood that training is not affected by quantization, hence I don't get "quantization-aware training".

The confusion is aggravated by "We train a three-layer convolutional neural network for all Atari games for 10 million steps, with a
quant delay of 5M (quantization aware training starts at 5M steps)." Why mention QAT in a section devoted to PTQ? I think I'm missing something about the methods that the authors did not properly explain. Based on the many typos in these sections (see below), I suspect they have been reorganized at the last minute and confusions have been introduced.

In PTQ, it is not clearly stated where dequantization is applied and whether it is applied in the real RL method or just for the sake of temporarily evaluating the method ("to simulate quantization error"). An explanation of your intent is missing here.

The quantization delay of 5M is mentioned but not explained nor justified. Why don't you start quantizing immediately? This must be explained.

The text about Table 3 mentions 8-bit and 16-bit precision, but in the table we can only see int8 and fp16. Something is inconsistent here.

In Table 3, you may add the relative impact (e.g. -X% or + X%) and put the most important ones in bold or color to highlight them.

Section 4 describes the experimental setup, but Section 3.2 already contains experimental studies and results. I suggest starting the experimental section at 3.2 and asking a list of addressed questions: the first corresponding to Section 3.2 and the others to the rest of experimental questions.

p3:
"reinforcement learning poses a unique opportunity" an opportunity for what? The claim is unclear.

In Table 4, the "difficulty" statement is quite subjective (though it comes from another author). By the way, why use "N/A" for the last three rather than "trivial" or "easy". I suggest removing this column or replacing it with something more objective (dimensionality, number of steps for SOTA convergence...?).

Why not use the same environments and algorithms in Section 3.2 and Section 4?

The caption of Table 5 says: "8 bit inference yields > 1.5 × −2.5× speedup over full precision training" but the column shows speedup between x1.51 and x5.41. Which is right?

As RL systems are scaled to run on 1000’s distributed CPU cores and accelerators (GPU/TPU), the carbon reduction can be significant.
=> well, the rate is still the same, and there is no reduction on the GPU/TPU part, right?

Why not run the experiments of Fig. 9 on Humanoid, as in Fig. 8, where the phenomenon was assessed?

The difference in impact between Figs 9a and 9b is far from obvious, so if there is such a difference, it should be better outlined with a more appropriate graph.

The relationship between Fig. 2 and Fig. 10 needs to be better outlined. Is it just the "broadcast" part that is scrutinized?

------------------------------

Minor and/or local issues, typos:

Abstract: > 1.5x - 2.5x is hard to parse. I could not make sense out of this before reading "between 1.5x and 2.5x" later on. This second formulation is clearer.

"versus reinforcement learning in various tasks. Across various settings, we show..."
=> versus reinforcement learning *algorithms* in various tasks and settings, We show...

By the way, you may introduce "RL" early in the text and use it consistently.

Figure 2, which is referred to in page 2, should be moved more forward (page 1 or 2).

the the overhead

thus paving way for => thus paving the way towards

"Quantization as applied to reinforcement learning has been relatively absent in the literature."
=> remove "relatively" to be consistent with a previous claim

Deepmind Acme => consistently use upper or lowercase. I suggest using \sc (small capitals) for all algorithms/frameworks.

thus paving the path for => thus paving the way towards (there are a few repetition which should be avoided)

actors, who then uses => use

"This is unlike traditional quantized neural network training, which must utilize more complex algorithms like loss scaling Das et al. (2018), specialized numerical representations Sun et al. (2019); Wang et al. (2018), stochastic rounding Wang et al. (2018) to attain convergence."
=> all refs should use \citep instead of \cite here.

"This adds extra complexity and may also limit speedup and, in many cases, are still limited"
=> and, in many cases, this is still limited...

the labels "fp16, fp32, int8..." should be explained at least once: floating point precision with 16 or 32 bits, or binary integers on 8 bits, etc.

We apply the PTQ to Atari arcade learning (Bellemare et al., 2012), OpenAI gym environments (Brockman et al., 2016b) and different RL algorithms namely A2C (Mnih et al., 2016), DQN (Mnih et al., 2013a), PPO (Schulman et al., 2017), and DDPG (Lillicrap et al., 2015) and.
=> remove final "and" (or complete the sentence)

Table 3 shows the rewards attained by policies quantized via post-training quantization in.
=> in what???

The plot shows that the 8-bit quantization can sufficiently capture the entire weight distribution policy trained by PPO and A2C. Whereas for DQN, ...
=> Whereas should not start a sentence

a gaussian noise layer => Gaussian

end to end => end-to-end (several times)

with a 2.8× reduction in carbon emissions.
=> with an average 2.8× reduction in carbon emissions.

(e.g., distillation, sparsification, etc). => etc.). (missing dot)

**Strengths And Weaknesses:**

Strengths:
- the idea of using quantization in RL is novel
- using quantization is shown to reduce carbon emission, whereas RL carbon emission  should be a growing concern. Additionally, it is shown to speed up learning, which is more a nice side effect than the primary concern

Weaknesses:
- parts of the methods and experimental setup are unclear.
- the empirical study has several flaws (see below)
- the writing needs significant improvement, with several unclarities and many typos that could easily be avoided. Most of this is easy to fix.

---

> ### Author Response · Authors · 2022-05-06
> **Reply to Reviewer nimW**
>
> We thank the reviewer for detailed feedback. We will make the necessary changes requested by the reviewer. TMLR guidelines ask authors to wait for all reviews before making changes to the paper.
>
> **R: Consistent use of actors, learners, and broadcaster**
> * We will make the necessary modifications in Fig 1 to make distinction between “actors”, “learners”, and “broadcasters”. The quantizer is the stage that performs the quantization of the policy.
> * We perform this step before broadcasting the policy to all actors from learners to reduce communication cost. Once the actors receive the quantized policy, it uses quantized policy inference (saves computation cost).
>
> **R: Distinction between PTQ and QAT.**
> * We will add clarifications and necessary citations in the revised version.
> * The “Quantizer” block in Fig 1 can be realized in many ways. The two popular forms of quantization (extended from supervised learning) are PTQ and QAT.
> * Before we design the quantizer block, we wanted to answer the following questions:
>   * *Does PTQ negatively affect the agent’s reward?* This is important because RL has an inherent feedback loop between agents and their environment. If the quantized policy introduces errors, we wanted to see if this error compounds due to the feedback loop. Our observation is that policies can be quantized up to 8 bits. No prior work was performed to answer such questions in the context of RL, hence the investigation was necessary as a first step. Thus, we included it.
>   * *How aggressively can we quantize?* QAT provides ways to quantize more aggressively than PTQ. Section 3.2 shows we can safely quantize the policy to up to 4-5 bits in most cases.
> *  As hardware architectures evolve to support native quantization below 8 bits (https://developer.nvidia.com/blog/int4-for-ai-inference/), the quantizer block can be modified to leverage those features.
>
> **R: Why is the quantization delay of 5M?**
> * Quantization is inherently a lossy operation. In QAT, a fake quantization node is inserted in the neural network graph, and the statistics of weights distribution are collected before the weights can be quantized.
> * The quantization delay is a hyperparameter that controls when the quantization is applied to the weights. One can consider this the “warm-up” period to collect the policy weight distribution statistics before applying quantization. We tried different values of quantization delay and used 5M since that gave us the best results.
> * We will clarify this in our revision and include a brief discussion.
>
> **R: Inconsistent use of 8-bit/int8 and 16-bit/fp16**
> * We will clarify this in our revised version and will use the int8 and fp16 throughout the paper for consistency.
>
> **R: Use +x%, bold or color to highlight them.**
> * This is a great suggestion and we agree that it will improve the readability of the table and focus on key trends. We will be happy to make these changes in our revision.
>
> **R: Replace subjective metrics (e.g., difficulty) with more objective (dimensionality, number of steps for SOTA convergence)**
> * We agree that the definition of “trivial”, “easy” was defined in another paper and we choose to use that nomenclature in our work.
> * For the columns with NA, these are stock openAI gym environments and no such definitions were provided for these environments.
> * We will remove this nomenclature and add more objective metrics in Table 4.
>
> **R: 8 bit inference yields > 1.5 × −2.5× speedup over full precision training" but the column shows speedup between x1.51 and x5.41. Which is right?**
> * Both are right. In the caption, we claim > than 1.5x to 2.5x. We will fix the upper bound gains based on the numbers in the column.
>
> **R: Similar reuduction rate as RL scales to 1000’s of CPUs?**
> * We would like to clarify that we were referring to reduction in absolute numbers when talking about running RL at scale (1000’s of distributed CPUs). A reduction from 2 Kgs of CO2 to 1 Kgs of CO2 is 50%. Likewise a reduction from 2000 KGs of CO2 to 1000 KGs of CO2 is also 50%. But in terms of absolute numbers, 1000 KGs savings is quite significant. We will clarify this in our revision.
>
> **R: Why not run the experiments of Fig. 9 on Humanoid, as in Fig. 8, where the phenomenon was assessed?**
> * We will update Fig 9 with Humanoid instead of Walker Stand. We believe the underlying tradeoff holds true irrespective of the environment.
>
> **R: The relationship between Fig. 2 and Fig. 10 needs to be better outlined. Is it just the "broadcast" part that is scrutinized?**
> * Fig 2 motivates why we focus on quantizing actors and not learners.  Fig 10 shows that quantization plays a role for both computation heavy scenarios as well as communication heavy scenarios (where the policy needs to be broadcasted often).
>
> **R: Other typos/local issues/restructure results section**
> * We thank the reviewer for in depth feedback on the typos and suggestions. We will carefully proofread and fix these issues in the next revisions.

---

> > ### Comment · Reviewer_nimW · 2022-05-20
> > **Update submission ?**
> >
> > Now that all reviews are present and answered, shouldn't the authors post a revised version of their submission?

---

> > > ### Author Response · Authors · 2022-05-20
> > > **Regarding Paper revisions**
> > >
> > > Thanks for the question. We are in the process of updating the paper with results from KL-divergence, DQN scaling experiments and writing changes requested by all the reviewers. We plan to submit the revised version by tomorrow (05/21).

---

> > ### Comment · Reviewer_nimW · 2022-05-29
> > **Comments on the revised version**
> >
> > I have read all the discussion threads and all the changes made by the authors in the revised version of the paper.
> >
> > My general feeling is that the paper was significantly improved. In particular, focusing on PQT is a good choice, and most of my previous concerns have been addressed. A remaining concern is that the paper is growing fast (from 19 to 24 pages) so the authors should start being careful about the paper length, but up to now it is not unreasonable. Below a list of typos and minor points:
> >
> > - many times, the authors write "between Ax to Bx". It should be either "between Ax and Bx", or "from Ax to Bx".
> > - pp. 7-8, my feeling is that some remarks about the multiagent case and using importance weighting are hardly relevant. I would suggest the authors to shorten this and stick to their own line of "story telling".
> >
> > p4: "and finally, learning a neural network policy based on the experience generation." I would remove "neural network" as you are speaking of RL in general, and RL does not imply using neural network policies.
> >
> > In 4.2 you should explain how you compute the carbon emission ratio.
> >
> > typos:
> >
> > p2: "Ape-X (Horgan et al., 2018), ACME (Hoffman et al., 2020b)." => missing a "or" or an "and" between both algos.
> >
> > p4: "as actor- learner training paradigm" => as an actor...
> >
> > same line: (Horgan et al., 2018)as => missing space
> >
> > p5: the leaner => learner (use a spell checker on your source)
> >
> > p7: The small change in action distribution for quantized policy suggest => suggests
> >
> > p8: (See Appendix A ) => (See Appendix A) (remove space)
> >
> > (>1.5 ×-5.41 ×) => between 1.5× and 5.41×
> >
> > p10 (caption) "> 1.5 × to 5.41 ×" => see above
> >
> > p13 "since there is less communication (Update Freq3̄00)" => write it properly...
> >
> > The ref about seed RL (Lasse Espeholt, Raphaël Marinier,...) is duplicated. Same about ACME (Matt Hoffman, Bobak Shahriari...), OpenAI gym (Greg Brockman, Vicki Cheung...), ALE (M. G. Bellemare, Y. Naddaf,...), "Quantizing deep convolutional networks" (Raghuraman Krishnamoorthi), DQN (Volodymyr Mnih, Koray Kavukcuoglu...), Ray (Philipp Moritz, Robert Nishihara...), Adaptivefloat (Thierry Tambe, En-Yu Yang...).
> > This will save a whole page ! ;)

---

> > > ### Author Response · Authors · 2022-05-30
> > > **Response to recent suggestions**
> > >
> > > We thank the reviewer for acknowledging that we were able to address most of their concerns. We also thank the reviewer for more suggestions on improving the quality of the presentation and the manuscript.
> > >
> > > We have updated the manuscript to include the suggestions and other typos in the previous version based on the recent comments. We have uploaded the revised manuscript with these changes.
> > >
> > > Currently we were able to reduce the page count to 22 pages (from 24 pages). Please let us know if there is a target page count to meet. We will try our best to tighten it to meet that target page count.

---

### Review · Reviewer_3T9V · 2022-05-12

**Summary Of Contributions:**

The paper presents an actor-network quantization method, ActorQ, to speed up data collection in distributed RL training. Specifically, this work adds a Quantizer between Actors and Learners to perform rollouts. The Quantizer is an inference-only actor and enjoys the usage of low precision operators to speed up training.

**Requested Changes:**

1. Adding a clarification of the consequent consequence after using the Quantizer in on-policy RL algorithms.

2. Adding a metric to measure the policy shift caused by quantized policy and discuss how it would affect the RL algorithm performance.

**Strengths And Weaknesses:**

Strengths

Overall, it is a good empirical paper which provides new insight into low-cost and high-efficiency experience collection. The experimental results have demonstrated good speedup and convergence performance in common RL tasks.

Weaknesses

1. Effects of quantization are unclear. This paper adopts two types of quantization: post-training quantization (PTQ) and quantization aware training (QAT). Essentially, the quantized policy is an actor policy close to the full-precision policy. Although the authors argue that the difference is small enough to maintain a good policy and large enough to regularize model behaviour. Therefore, it is worth using some quantifiable metrics (e.g., KL divergence) to measure the distribution shift between quantized and full-precision policies.

2. When using the Quantizer for collection experience, the policy shift mentioned above would break the on-policy assumption in some RL algorithms(e.g., PPO). However, it is unclear how to deal with this problem (correct me if I have missed something).

3. In some distributed RL frameworks, like SeedRL, the policy training and inference are both on GPU. In this case, the policy inference can also be sped up, and the communication can be reduced. So it would be good to clarify if the proposed quantized can also be used in the frameworks like SeedRL.

---

> ### Author Response · Authors · 2022-05-19
> **Reply to Reviewer 3T9V**
>
> We thank the reviewer for the feedback and acknowledge that this is an empirical paper.
>
> **[R]: Effects of quantization are unclear. Please use KL-divergence to quantify the distribution shift.**
>
> * We thank the reviewers' suggestions to use KL-divergence to understand the effects of quantization better.
> The table below captures the KL-divergence between the action distribution for the fp32 policy and int8 policy for PPO, A2C, DDPG, and DQN.
> | **Algorithm (&rarr;)**   | **PPO**    | **A2C**    | **DDPG**   | **DQN**    |
> |--------------------------|------------|------------|------------|------------|
> | **Environment** (&darr;) | **KL-Div** | **KL-Div** | **KL-Div** | **KL-Div** |
> | **Breakout**             |    0.08892 |    0.00262 |            |    0.06045 |
> | **SpaceInvaders**        |    0.08115 |    0.06066 |            |    0.01353 |
> | **BeamRider**            |     0.0186 |    0.00993 |            |    0.09787 |
> | **MsPacman**             |    0.11978 |    0.17536 |            |    0.01531 |
> | **Qbert**                |    0.02573 |    0.01957 |            |    0.01995 |
> | **Seaquest**             |    0.02991 |    0.03358 |            |    0.02536 |
> | **CartPole**             |    0.00566 |    0.00113 |            |     0.1019 |
> | **Pong**                 |       0.01 |    0.01528 |            |    0.12572 |
> | **Walker2D**             |       0.03 |    0.05371 |     0.0376 |            |
> | **HalfCheetah**          |       0.02 |    0.13427 |     0.0763 |            |
> | **BipedalWalker**        |       0.03 |     0.0252 |     0.0134 |            |
> | **MountainCar**          |       0.07 |    0.03705 |     0.0651 |            |
>
>
> * We observe that the KL-divergence is very small, and visualize the action distribution for several algorithm/environment combinations. The plots, which will be included in the updated manuscript, are available here: https://bit.ly/action_dist_plots
>
> * We would like to clarify that PTQ and QAT are studies undertaken to evaluate the effects of two different ways to do quantization in RL.  PTQ is the simpler form of quantization and is widely supported. QAT allows quantization to fewer than one byte (8 bits). The QAT study assessed how aggressive we can quantize the actor’s policy, if hardware and libraries that support sub-byte quantization are widely available.
> * The ActorQ uses only PTQ, and not QAT, since we have policies that can be quantized safely with 8-bit precision and all common hardware (CPUs, GPUs, and TPUs) provides native support for int8 computations. For better readability, we will move our QAT results to the appendix since the Quantizer block in ActorQ only uses PTQ.
>
> **[R]: Clarify about Quantizer effects on-policy RL algorithms.**
> * For the two on-policy algorithms we evaluated, PPO and A2C, the KL-divergence between the fp32 and quantization policy is minimal suggesting that the policy shift is not too significant. This is also reflected in the agent’s reward where the quantized policy achieves similar comparable reward to fp32 policy.
>
> * Recent progress in RL/MARL shows that on-policy algorithms like PPO adapts well to small distributional shifts in the environment. For instance, in MARL, the non-stationarity problem is caused by other agents’ policy change during learning and independent PPOs surprisingly perform better than centralized on-policy RL algorithms (https://arxiv.org/abs/2011.09533, https://arxiv.org/abs/2103.01955). The on-policy RL algorithm performance does start to break down once the distribution shift becomes too large (e.g. agents behaviors are too different: https://arxiv.org/abs/2010.00581, https://arxiv.org/abs/2201.08896 ). Given that in our case the KL divergence is small, indicating that the distribution shift between learner and actor policies are small enough to stay within the acceptable range. We hypothesize that even though it is technically adding quantization to the on-policy RL results in off-policy learning, that the distribution shift caused by quantizing the actor policy is similar in nature to the multi-agent setting with the independent learning, and the distribution shift caused there was handled well by PPO.
>
> * One possible way to counteract large policy shifts, if they occur and the training is no longer effective, is to use importance weighting (https://arxiv.org/pdf/1802.01561.pdf) to correct the discrepancy between two policies. We will be happy to add a discussion in our paper.
>
> **[R]:Regarding SeedRL**
> * Most popular hardware accelerators (GPUs/TPUs) provide native hardware support for quantization. Quantized inference has shown benefits in speeding up the inference while lowering energy consumption.
> * We believe that quantization can also be applied to SeedRL and have the following benefits:
>    * Speed-up in actor’s inference with quantized computation.
>    * Lower the data transfer cost from DRAM/caches to accelerator. 8-bit quant.ized policy can save memory footprint by 4x compared to fp32 policy.

---

### Author Response · Authors · 2022-05-21
**Summary of Revisions**

We thank the reviewers for their detailed comments and suggestions to improve the manuscript. We have uploaded a revised version based on the feedback received so far. The revisions are marked in blue text. We also summarize the changes made to the paper.

**Summary of revisions**

 * **KL-Divergence metrics**
   * We updated table 2 with the KL divergence metric between the fp32 policy and int8 policy for DQN, PPO, A2C, and DDPG for several   RL tasks.
   * Visualization of action distribution between fp32 policy and int8 policy
   * Added Fig 3, which visualizes the action distribution between fp32 policy and int8 policy with two on-policy algorithms, namely A2C and PPO, for the WalkerStand environment. Appendix C has plots for other environments.
* **Confusion due to PTQ and QAT**
  * We have moved the QAT to the appendix to focus on PTQ, which is used in ActorQ quantizer in this work. So we focus on PTQ in the main manuscript.
  * QAT is another way to perform quantization which allows for aggressive quantization (i.e., below 8-bits). As native sub-byte quantization support in hardware and optimization libraries become more prevalent, the quantizer block in ActorQ can leverage more aggressive forms of quantization.
 * **Focus on distributed RL training in Table 1**

   * We have added a scaling study to show the benefits of quantization for both distributed and non-distributed RL training scenarios: # of actors = 1 is the non-distributed scenario and # of actors >1 is the distributed scenario.

   * Fig 11 (Appendix B) plots the number of actors on x-axis and wall clock time on y-axis for DQN agent. We scale the number of actors from 1 to 8. We see a linear saving O(n) as the number of actors increases. We also see that quantization can benefit the non-distributed scenario since the inference cost ($c_{a}$) is lowered.

* **Other minor writing changes, typos, and figures.**
  * We have fixed most of the typos highlighted by “reviewer nimw” and “reviewer axpH.”
  * Fig 2: We also add a subfigure to show the traditional RL training setup next to ActorQ.
  * Fig 7(b) adds Walker stand and the Humanoid stand environment to be consistent with Fig 8. We choose to add WalkerStand since it reduces the number of experiments we need to run. However, we believe that the trend should hold for other environments evaluated in ActorQ.
  * We moved Fig 1 to page 2 (where the figure was discussed)
  * We dropped the “Difficulty” header from Table 3. We strongly believe that we need a leaderboard for all the environments which compares in a fair way across various algorithms. Since this is not the focus of our work, we decided to drop the difficulty header. Please let us know if you strongly feel it should be included. We will be happy to do a more in-depth literature review to see if we can find a prior work that provides an acceptable taxonomy on task difficulty.
  * Added discussion on KL-divergence, policy shift, and its impact on on-policy algorithms.
  * Consistent reporting of the speed-up number.

---

### Author Response · Authors · 2022-05-29
**Summary of revisions**

We thank the reviewer axpH for the clarification questions in the latest response. To that end, we have updated the paper to provide more clarification. Please let us know if these changes alleviate your concerns.

Here is the summary of modifications made in this revisions

**Abstract and Introduction**
* Abstract now includes a range for both speedups and carbon emission reduction as originally reported in Table 4.
* We have added clarifications in the introduction section to make distinction between the ActorQ and benchmarking study. We hope this distinction helps in answering reviewer axpH  questions on why we use different algorithms in ActorQ and benchmarking studies.

**Table 4 carbon emission reporting as ratios of two quantities**
* We have replaced the % improvement in Table 4 to ratio of carbon emissions between fp32 policy and int8 policy. Here is the reasoning

*  % improvment from “A” and “B” is defined as:
   $$ (A-B)*100 \over A$$

* We choose fp32 CO2 emissions as “A” and int8 CO2 emissions as “B”. This was an arbitrary choice and alternatively one can flip the ordering between these two and arrive at a different % change number.

* Hence, similar to how we report speedup number, we also report the carbon emissions. We define the ratio as in Table 4:
   $$ CO2_{fp32} \over CO2_{int8}  $$

 Where,

CO2$_{fp32}$: Absolute carbon emission numbers (Kgs) when using fp32 policy

CO2$_{int8}$: Absolute carbon emission numbers (Kgs) when using int8 policy

---

### Author Response · Authors · 2022-06-08
**Follow up**

We would once again like to thank you all the reviewers for providing valuable feedback/suggestions for improving the manuscript so far. Kindly let us know if you have any questions still not addressed. We would be happy to address them or make more revisions to the paper.

---

### Decision · Action_Editors · 2022-06-20

**Recommendation:** Accept as is

**Comment:**

The paper describes a new framework to speed up RL techniques by quantization.  This work makes an important contribution to the literature of quantization for RL as well as sustainable RL.  It includes a good empirical analysis that clearly shows the benefits of the framework.  While the paper was initially unclear, the authors made many revisions that addressed the concerns of the reviewers.  The paper is now ready for publication.